# Type IVb Hypersensitivity Reaction in the Novel Murine Model of Palladium–Induced Intraoral Allergic Contact Mucositis

**DOI:** 10.3390/ijms24043137

**Published:** 2023-02-05

**Authors:** Keisuke Nasu, Kenichi Kumagai, Takamasa Yoshizawa, Kazutaka Kitaura, Ryota Matsubara, Motoaki Suzuki, Ryuji Suzuki, Yoshiki Hamada

**Affiliations:** 1Department of Oral and Maxillofacial Surgery, School of Dental Medicine, Tsurumi University, Yokohama 230-8501, Japan; 2Department of Rheumatology and Clinical Immunology, Clinical Research Center for Rheumatology and Allergy, Sagamihara National Hospital, National Hospital Organization, Sagamihara 252-0392, Japan; 3Department of Oral and Maxillofacial Surgery, Dentistry and Orthodontics, The University of Tokyo Hospital, Tokyo 113-8655, Japan; 4Repertoire Genesis Inc., Osaka 567-0085, Japan; 5Department of Oral and Maxillofacial Surgery, Sendai Tokushukai Hospital, Sendai 981-3116, Japan; 6Department of Anatomy and Physiology, Faculty of Medicine, Saga University, Saga 849-8501, Japan

**Keywords:** palladium allergy, contact dermatitis, allergic contact mucositis, metal allergy

## Abstract

Palladium (Pd) is a component of several alloy types that are widely used in our environment, including several dental alloy types that cause adverse reactions such as hypersensitivity in the oral mucosa. However, the pathological mechanism of intraoral Pd allergies remains unclear because its animal model in the oral mucosa has not been established. In this study, we established a novel murine model of Pd–induced allergies in the oral mucosa, and explored the immune response of cytokine profiles and T cell diversity in terms of the T cell receptor. The Pd–induced allergy mouse was generated by two sensitizations with PdCl_2_, plus a lipopolysaccharide solution into the postauricular skin followed by a single Pd challenge of the buccal mucosa. Significant swelling and pathological features were histologically evident at five days after the challenge, and CD4–positive T cells producing high levels of T helper 2 type cytokines had accumulated in the allergic oral mucosa. Characterization of the T cell receptor repertoire in Palladium allergic mice indicated that Pd–specific T cell populations were limited in V and J genes but were diverse at the clonal level. Our model demonstrated that a Pd–specific T cell population with Th2 type response tendencies may be involved in the Pd–induced intraoral metal contact allergy.

## 1. Introduction

Metals are ubiquitous in our environment, and often cause allergic contact dermatitis (ACD), which is an inflammatory disease categorized as a delayed–type hypersensitivity (DTH) reaction [1]. Epidemiological studies reported palladium (Pd) as the hapten with the highest prevalence in ACD pathogenesis after nickel (Ni) [2,3]. Pd has a close chemical resemblance with Ni and platinum (Pt), and its main applications include the crown and bridgework in dentistry and is sometimes used in jewelry. Pd allergy incidents have recently increased in patients with dermatitis and dental disorders [4].

Most dental appliances to restore or replace decayed teeth are partially composed of alloys that may contain a large variety of metals. Corrosion is an inevitable chemical reaction between the oral environment and dental alloys that may lead to substantial and clinically relevant ion release and in turn, result in adverse reactions, such as hypersensitivity and ACD. A strong relationship was found between the exposure to dental alloys and ACD; thus, dental crowns seem to play a key role [5,6]. However, the relationship between hypersensitivity to metals and objective oral abnormalities is unclear. Most likely, the ability of various metals to trigger the innate immune system and the tolerant character of the oral mucosa (OM) play a key role.

The DTH immune response in the OM differs from that in the skin mainly due to differences in the accumulation of local antigen–presenting cells and T cells [7,8]. Previous studies reported that DTH reactions in the oral mucosa have the hallmarks of skin DTH reactions with T cells and macrophages [9]. The involvement of pathogenic T cells in the development of a metal allergy in the oral environment has not yet been explored using animal models, although metal allergies have long been known to be T cell–dependent and the metal ions caused by the corrosion from dental alloys are known to function as haptens. We recently generated the novel murine model of Ni–induced allergic contact mucosa (ACM) and have analyzed antigen–specific immune responses [10]. However, the conformation and toxicity of Ni and Pd were different, and the mechanism of the specific immune response to Pd in the OM remained unknown.

Recent advances in next–generation sequencing (NGS) have enabled massive sequencing and long–read sequencing, and the development of NGS–compatible bioinformatics software has enabled NGS–based T cell receptor (TCR) repertoire analysis, which is a quantitative and comprehensive analysis of a large–scale repertoire [11,12,13].

In the present study, we generated a novel murine model of Pd–induced allergies in the OM to explore how accumulated antigen–specific T cells at the site of allergic inflammation contribute to Pd allergy development in the OM.

## 2. Results

### 2.1. Pd–Induced Allergic BALB/cAJcl Mice Develop Swelling in the OM

The swelling of the buccal area of the OM of ACM mice reached a maximum on day 2 after the challenge (Figure 1A). Significant redness and swelling are evident in the buccal area of ACM mice compared with the control and ICM mice on day 2 after the challenge (Figure 1B). Conversely, OM buccal area swelling in irritant contact mucositis (ICM) and control mice reached a maximum on day 1 after the challenge. The difference in buccal mucosal swelling was significantly higher in ACM mice compared with ICM mice on days 2, 3, and 5 after the challenge. The most differences in OM buccal area swelling between ACM and ICM were observed in day 2 after the challenge. We focused our analysis on day 2, when ACM had the largest OM swelling and the most significant difference between ACM and ICM, and day 5, when ACM had significantly larger OM swelling than ICM and the largest CD3 mRNA expression levels in OM.

### 2.2. Histopathological and Immunohistochemical (IHC) Analyses of F4/80 and CD3 in the OM of Pd–Induced Allergic BALB/cAJcl Mice

We examined the histopathological and IHC analyses in the OM of control and Pd–induced ACM and ICM mice on days 2 and 5 after the challenge to verify whether macrophages and T cells infiltrated into the inflamed OM. Hematoxylin and eosin (HE) staining showed the dense infiltration of inflammatory cells in the epithelial basal layer and upper dermis, as well as swelling of the OM epithelium and epidermal spongiosis in ACM mice on day 5 after the challenge, but not in ICM and control mice (Figure 2A,C,E).

Next, we performed an IHC analysis of F4/80 and CD3 in the OM of control and Pd–induced ICM and ACM mice to verify whether macrophages and T cells had infiltrated into the inflamed OM of ACM mice (Figure 2F–O). IHC staining showed considerable F4/80+ macrophage infiltration into the dermis of ICM and ACM mice on day 2 after the challenge compared with control mice (Figure 2F,G,I). Moreover, the F4/80 staining intensity was stronger in ACM than in ICM mice. In contrast, F4/80+ cells were lost on day 5 after the challenge, both in ACM and ICM mice. IHC analyses of CD3+ T cells were observed in the epithelial basal layer and upper dermis in the ACM mice, and they were continued between day 2 and day 5 after the challenge (Figure 2N,O). Conversely, CD3+ T cells were slightly observed in ICM mice (Figure 2L,M).

### 2.3. mRNA Expression Levels of T Cell Markers in the OM of Pd–Induced Allergic BALB/cAJcl Mice

We performed a quantitative polymerase chain reaction (qPCR) analysis in CD3, CD4, and CD8 expressions to verify whether mRNA expression levels of T cells infiltrated into the inflamed OM. The CD3 expression ratio was significantly increased in control ACM mice compared to ICM mice on days 2 and 5 after the challenge (Figure 3). Furthermore, the ratio of CD3 expression in ACM tended to increase from day 2 to day 5. A similar expression trend to CD3 was observed for CD4, which was significantly increased in ACM mice compared to ICM mice. Conversely, no differences were found in the CD8 expression ratio between ACM and ICM mice.

### 2.4. mRNA Expression Levels of T Cell–Related Cytokines in the OM of Pd–Induced Allergic BALB/cAJcl Mice

We compared the expression levels of Th1–related cytokines (interleukin (IL)–2, interferons (IFN)–γ, and tumor necrosis factors (TNF)–α), and Th2–related cytokines (IL–4 and IL–10), in the OM of Pd–induced ICM and ACM mice to examine inflammation in allergic OM (Figure 4). The expression level of Th2–type cytokines was higher in ACM mice compared with ICM mice from day 2 to day 5 after the challenge. A large difference in the expression levels of ACM and ICM mice was observed, especially in IL–4, and the difference in IL–10 between ACM and ICM mice appears to be larger on day 5 than on day 2 after the challenge. Conversely, the expression levels of TH1–related cytokines were closely similar between ICM and ACM mice, except for IL–2 on day 2 after the challenge.

### 2.5. TRV–TRJ Combination and Diversity of TCR Repertoire in the OM and Cervical Lymph Nodes (Ly) of Pd–Induced ICM and ACM Mice on Day 5 after the Challenge

We analyzed the TRV and TRJ expression levels in the inflamed OM and Ly by a NGS–based TCR repertoire analysis to determine the TCR repertoire of T cells in a Pd allergy that had infiltrated into the OM and Ly of ICM and ACM mice on day 5 after the challenge. A representative TRV–TRJ combination three–dimensional graph of the TRA and TRB repertoire in OM revealed the dominance of certain TRV and TRJ gene combinations as well as the extent of the diversity of TCR usage (Figure 5A). Hence, more TRV–TRJ combinations of TRA and TRB repertoire were detected in ACM OM than in ICM OM mice. Next, we investigated the diversity of repertoire in TRA and TRB using the Shannon index in ACM and ICM samples (Figure 5B). The Shannon index was significantly higher in ACM OM than in ICM OM mice. Furthermore, ACM Ly showed a trend toward a higher Shannon index than ICM Ly mice.

### 2.6. Commonality of TCR Repertoire in the OM and Ly of Pd–Induced ICM and ACM Mice on Day 5 after the Challenge

We compared the top 20 in the ACM OM mice to other samples to assess the commonality of unique reads obtained from the repertoire analysis in TRA and TRB (Figure 6A,B). The top reads comprising the ACM OM mice shared many with ACM Ly (green shading) and some with ICM Ly, but not with ICM OM. The TRA sequences (TRAV11D, TRAJ18, and CVVGDRGSALGRLHF) of iNKT cell (orange shading) were detected in all samples, and the frequency was lower in ACM OM (0.02%) than in ICM OM (1.12%) mice (Figure 6B).

### 2.7. Evaluation of TRV and TRJ Gene Skew in Common Reads of ACM OM and ACM Ly Repertoire

The commonality was assessed for all unique reads in the ACM OM mice for further investigation (Figure 7A,B). Of the 12,608 TRA unique reads of ACM OM mice, 829 reads with ACM Ly, 494 reads with ICM Ly, and 19 reads with ICM OM mice were shared. Of the 15,917 TRB unique reads of ACM OM mice, 630 reads with ACM Ly, 186 reads with ICM Ly, and 16 reads with ICM OM mice were shared.

The number of unique reads present in ACM OM to those that were increased in ACM Ly mice over ICM were 683 and 550 for TRA and TRB, respectively. Furthermore, the reads were limited to 524 and 529 in TRA and TRB, respectively, of the unique reads held by ACM Oms mice, when narrowed down to those that were present only in ACM Ly mice (blue box line). 

The frequency counts of TRA and TRB, and TRV and TRJ gene usage were tabulated for the number of unique reads in the two narrowed–down patterns (ACM Ly over ICM, or only in ACM Ly) (Figure 7C–F). The genes in TRAV were distributed in 63 of 112 genes with a mean of 10.8 or 8.3 read counts, and the genes in TRAJ were distributed in 41 of 51 genes with a mean of 16.7 or 12.8 read counts. The genes in TRBV were distributed in 20 of 23 genes with a mean of 27.0 or 26.5 read counts, and the genes in TRBJ were distributed in 12 of 13 genes with a mean of 45.8 or 44.1 read counts. Genes detected at high frequency counts (mean + 2SD, beyond the green or blue line) were TRAV7D–2, TRAV14–1, TRAJ18, TRBV5, and TRBJ2–7.

## 3. Discussion

In the present study, we established a novel murine model for Pd–induced allergies in the OM in which CD3+ T cells infiltrated the inflamed area. Previous studies have reported a Pd–allergic model mouse in the footpad skin or auricle regions [14,15]. However, the pathological mechanism of the intraoral Pd allergy remains unclear because an animal model of Pd allergies in the OM has not been established. This is the first report to clarify Pd–specific immune responses in the OM using a murine model of intraoral Pd allergies.

OM swelling of ACM and ICM mice revealed that OM swelling could be caused by nonspecific stimulation to external stimuli only, as shown by the swelling on day 1 without significant difference. However, on day 2, the OM buccal swelling in ICM mice significantly decreased, with the largest significant difference between ACM and ICM mice. Thereafter, significant differences between ACM and ICM mice were observed until day 5, but not on day 7. These findings suggest that inflammation other than nonspecific stimulation to external stimuli is induced in ACM mice [16].

Therefore, we confirmed the histopathological and IHC analyses in ACM and ICM mice from days 2 and 5 after the challenge and control mice. HE staining revealed the dense infiltration of inflammatory cells in the epithelial basal layer and upper dermis, as well as swelling of the OM epithelium and epidermal spongiosis in ACM mice on day 5 after the challenge that was more markedly than others. Spongiform edema in the epithelium and swelling of the epithelium are characteristics of delayed allergy [17,18]. Marked infiltration of F4/80+ macrophages and CD3+ T cells were found in ACM mice on day 2 after the challenge, and CD3+ T cells increased on day 5 from day 2 after the challenge. This innate works in both ACM and ICM mice on day 2 by invading the PdCl_2_ solution in vivo. Additionally, on day 2, ACM mice admitted CD3 and F4/80 macrophages more remarkably than ICM mice, suggesting that adaptive immunity works in ACM mice. CD3+ T cell infiltration, mucosal epithelial thickening, and supratentorial edema increased from day 2 to day 5 in ACM mice. Our results suggested that allergic inflammation in the OM was initiated by the response of macrophages to Pd, followed by T cell infiltration into the inflamed oral mucosa after antigen presentation by macrophages. This suggested that the ACM mice induced a metal allergy, which was stronger on day 5.

Four categories exist within type IV hypersensitivities, including type IVa, which is a CD4+ T helper (Th) 1 lymphocyte–mediated reaction with macrophage activation; type IVb, which is CD4+ Th2 lymphocyte–mediated with eosinophilic involvement; type IVc, which is cytotoxic CD8+ T lymphocyte–mediated with perforin–granzyme B involvement in apoptosis; and type IVd, which is T cell–driven neutrophilic inflammation [13,19]. A previous study suggested that a metal allergy is associated with either CD4 or CD8 T cell activation depending on the antigen processing pathway involvement [20,21]. The established Pd–induced allergic mouse model in the present study indicated predominant CD4+ T cell infiltration and Th2–type cytokine production in the OM. These results suggest that the established mouse model recapitulates the pathology corresponding to a type IVb delayed allergy.

Diversity was increased in the repertoire analysis of ACM mice compared to ICM mice, suggesting that allergic responses locally induce a variety of T cells and that the affiliated lymph nodes may be the source of these cells. The presence of common unique reads in ACM OM and Ly mice provides evidence for the involvement of the affiliated lymph nodes in the induction of locally generated specific T cells. Conversely, the reason for the partial sharing of ICM Ly unique reads in ACM OM mice may be due to the residual acute–phase clones observed in the early inflammation and dragged by the allergic reaction. Additionally, these unique reads are not detected in ICM OM mice, suggesting that they are lost locally in non–allergic reactions at the time of day 5 after the challenge. The unique reads shared by ACM OM and ACM Ly mice were diverse but biased toward specific TRVs or TRJs, suggesting that specific T cell induction in Pd allergies exists in a wide range.

iNKT cells are characterized by the expression of an invariant TRA encoded by TRAV11 (Vα14)–TRAJ18 (Jα18) in mice and TRAV10 (Vα24)–TRAJ18 (Jα18) in humans [22,23]. Previously, we identified iNKT cells in the lymphocytic infiltrates at a high frequency during the elicitation phase in Ni, Cr, and Ti allergies [24,25,26]. Additionally, iNKT cells and TRAV6–6–TRAJ57 bearing T cells have been implicated in the immune response to a Ni–induced OM metal contact allergy [10]. Interestingly, the reduced TRA sequence of iNKT cells in ACM OM mice indicates that they were pressured by these Pd allergy–specific clone populations and may not have a major role, unlike Ni–induced intraoral metal contact allergies.

In conclusion, we have established a model of an intraoral Pd–induced metal allergy, and revealed that Pd–specific T cell populations are limited to V or J gene usage. The application of the direct cloning of TCR genes from local sites of inflammation in this model will be a powerful tool for advancing our understanding of T cell–mediated immune disease in metal allergies, as well as providing new insights into antigen recognition by Pd–specific TCR in the OM.

## 4. Materials and Methods

### 4.1. Animals

BALB/cAJcl mice (4–week–old females) were purchased from CLEA Japan (CLEA Japan, Tokyo, Japan). All mice were in good health throughout the study and were given 1 week to acclimate to their surroundings before the study began. At the beginning of the experiment, 5–week–old BALB/cAJcl mice were used. All mice were kept in plastic cages (with a lid made of stainless–steel wire) with food and water available ad libitum. They were kept in our conventional animal facility with a temperature of 19–23 °C, a humidity of 30–70%, and a 12–h day/night cycle. All surgeries were performed using three different types of mixed anesthetics, and every effort was made to minimize animal suffering. Three different types of mixed anesthetic agents were used to sacrifice all mice by cervical dislocation to ensure death and prevent pain caused by tissue harvesting.

### 4.2. Reagents

PdCl_2_ (purity of >95%) was purchased from FUJIFILM Wako Pure Chemical Co., Ltd. (FUJIFILM Wako Pure Chemical Co., Ltd., Osaka, Japan). Lipopolysaccharide (LPS) from Escherichia coli (O55:B5) prepared by phenol–water extraction was purchased from Sigma–Aldrich (Sigma–Aldrich, St. Louis, MO, USA). The dissolution used PdCl_2_ plus LPS in sterile saline (Otsuka Normal Saline, Otsuka Pharmaceutical Factory, Inc., Tokushima, Japan).

### 4.3. Anesthetic Agents

The anesthetic was created by combining three medications. Medetomidine hydrochloride was purchased from Nippon Zenyaku Kogyo Co., Ltd. (Nippon Zenyaku Kogyo Co., Ltd., Fukushima, Japan), midazolam from Sandoz (Sandoz, Tokyo, Japan), and butorphanol tartrate from Meiji Seika Pharma Co., Ltd. (Meiji Seika Pharma Co., Ltd., Tokyo, Japan). These drugs were kept at room temperature (RT). We combined doses of 0.3 mg/kg of medetomidine hydrochloride, 4 mg/kg of midazolam, and 5 mg/kg of butorphanol tartrate. A previous study determined the concentration ratio of the three types of mixed anesthetic agents [27]. Typically, 25 mL of anesthetic agent is prepared by combining 0.75 mL of medetomidine hydrochloride, 2 mL of midazolam, 2.50 mL of butorphanol tartrate, and 19.75 mL of sterile saline. All agents were diluted in sterile saline and stored in the dark at 4 °C and sterile saline at 4 °C. The mice were administered a volume of 0.01 mL/g of body weight of the anesthetic mixture. Every mouse was intraperitoneally injected with the mixture of the three types of anesthetic agents.

### 4.4. Experimental Protocol of the Mouse Model of Pd–Induced Intraoral Metal Contact Allergy

An experimental protocol for the induction of the metal allergy in the OM was developed based on a previous protocol for metal allergy induction in footpad skin [10,15]. Mice were separated into three groups: ACM mice (*n* = 15), ICM mice (*n* = 15), and control mice (*n* = 15), with each group consisting of randomly chosen mice. All experiments were conducted in another room upon transfer from the animal holding area.

Sensitization. Mice were intradermally injected with 125 µL of 10 mM of PdCl_2_ and 10 μg/mL of LPS in sterile saline twice with a 7–day interval between injections. ACM was applied to the left and right postauricular skin of mice. Mice were first challenged 7 days after the second sensitization.

Challenge for elicitation. In the preliminary experiments, the concentrations of PdCl_2_ in 5 μM, 10 μM, and 30 μM were tested. In the 30 μM, almost all mice died. Based on the difference in cheek swelling and CD3 expression levels in the concentration of the 5 μM and 10 μM, the 10 μM were selected as the indicated concentration. The immune response was elicited using 25 µL of 10 mM of PdCl_2_ without LPS in sterile saline. ACM mice were challenged by submucosal injection in the left and right buccal regions of the OM. Submucosal injections were used to challenge non–sensitized ICM mice in the left and right buccal areas of the OM. Mice sensitized with PdCl_2_ plus LPS and then challenged with sterile saline were used as a control.

### 4.5. Measurement of OM Swelling

Swelling of the buccal area was measured before the challenge and at 1, 2, 3, 5, and 7 days after the first challenge using a Peacock Dial Thickness Gauge (Ozaki MFG Co., Ltd., Tokyo, Japan). All procedures on mice under anesthesia were performed by the same person.

### 4.6. Histological and IHC Analysis

Buccal OM samples were obtained from control mice and Pd–induced ICM and ACM mice for histological and IHC analyses. Furthermore, tissue samples were immersed in 4% paraformaldehyde–lysine–periodate for 48 h at 4 °C. Following a 10–min PBS wash, fixed tissues were soaked in 5% sucrose/PBS for 1 h at 4 °C, 15% sucrose/PBS for 3 h at 4 °C, and finally, 30% sucrose/PBS overnight at 4 °C. Tissue samples were snap–frozen by immersion in a mixture of acetone and dry ice in Tissue Mount (Chiba Medical, Saitama, Japan). Frozen sections were cut into 6–µm thick cryosections and air–dried on poly–L–lysine–coated glass slides. The HE stain was applied to cryosections for histological analysis. Cryosections were stained with anti–mouse F4/80 (1:1000; Cl–A3–1, Abcam, Cambridge, UK) and anti–mouse CD3 (1:500; SP7, Abcam, Cambridge, UK) monoclonal antibodies for IHC analysis. Mouse macrophage populations were detected in many buccal OM tissues using the F4/80 monoclonal antibody. The sections were incubated at RT for 30 min in PBS containing 5% normal goat rabbit serum, 0.025% Triton X–100 (FUJIFILM Wako Pure Chemical, Osaka, Japan), and 5% bovine serum albumin (Sigma–Aldrich St. Louis, MO, USA). Sections were incubated with primary mAbs for 1 h at RT. Intrinsic peroxidase was inhibited with 3% H_2_O_2_ in methanol after three 5 min washes with PBS. Tissue sections were washed twice and incubated for 1 h at RT with a secondary antibody (biotinylated goat anti–hamster immunoglobulin G or biotinylated rabbit anti–rat immunoglobulin G) after soaking in distilled water. The sections were treated with Vectastain ABC Reagent (Vector Laboratories, Burlingame, CA, USA) for 30 min at RT, followed by 3,3–diaminobenzidine staining (0.06% diaminobenzidine and 0.03% H_2_O_2_ in 0.1 M Tris–HCl, pH 7.6; FUJIFILM Wako Pure Chemical Co., Ltd., Osaka, Japan). Tissue sections were counterstained with hematoxylin to visualize cell nuclei.

### 4.7. RNA Extraction and cDNA Synthesis

Fresh OM specimens were obtained from each mouse and immediately soaked in RNAlater RNA Stabilization Reagent (Invitrogen, Carlsbad, CA, USA). Total RNA from the OM was extracted using the RNeasy Lipid Tissue Mini Kit (Qiagen) following the manufacturer’s instructions. Complementary DNA (cDNA) was synthesized from DNA–free RNA using the PrimeScript RT reagent Kit (Takara Bio, Tokyo, Japan) following the manufacturer’s instructions.

### 4.8. qPCR

The expression levels of immune response–related genes, including T cell–related CD antigens, cytokines, cytotoxic granule, and transcription factors of regulatory T cells, were measured by qPCR using the Bio–Rad CFX96 system (Bio–Rad, Hercules, CA, USA). Specific primers for GAPDH, CD3, CD4, CD8, IFNγ, TNFα, IL–2, IL–4, and IL–10 were described [11,28]. Freshly isolated total RNA from the OM and submandibular lymph node specimens were converted to cDNA. The PCR consisted of 5 μL of SsoFast EvaGreen Supermix (Bio–Rad), 3.5 μL of RNase/DNase–free water, 0.5 μL of 5 μM of primer mix, and 1 μL of cDNA in a final volume of 10 μL. Cycling conditions were as follows: 30 s at 95 °C followed by 50 cycles of 1 s at 95 °C and 5 s at 60 °C. A melting curve analysis was performed from 70 °C to 90 °C at the end of each program to confirm the homogeneity of the PCR products. All assays were repeated three times, and mean values were used to calculate gene expression levels. Five 10–fold serial dilutions of each standard transcript were used to determine the absolute quantification, specification, and amplification efficiency of each primer set. Standard transcripts were generated by the in vitro transcription of the corresponding PCR product in a plasmid. The nucleotide sequences were confirmed by DNA sequencing using the CEQ8000 Genetic Analysis System (Beckman Coulter, Fullerton, CA, USA). An Agilent DNA 7500 Kit in an Agilent 2100 Bioanalyzer (Agilent, Santa Clara, CA, USA) was used to validate their quality and concentration. GAPDH gene expression was used as an internal control.

### 4.9. TCR Repertoire Analysis

Total RNA was extracted from the OM and Ly of ICM and ACM mice on day 5 after the challenge and the OM of ACM mice on days 2 and 7 after the challenge. NGS was used to perform a TCR repertoire analysis developed by Repertoire Genesis Inc. (Repertoire Genesis Inc., Osaka, Japan [13]). An unbiased adapter–ligation PCR was executed according to the previous report [29]. Superscript III reverse transcriptase was utilized to convert total RNA to cDNA (Invitrogen). Afterward, double–stranded (ds) cDNA was synthesized, and an adapter was ligated to the 5′ end of the ds–cDNA before it was cut with the SphI restriction enzyme. PCR was performed with a P20EA adapter primer and a TCR α–chain constant region–specific primer (mCA1) for TCRα. The second PCR was conducted using the same PCR conditions and primers, including mCA2 and P20EA. Primers were utilized for TCRβ, mCB1, and mCB2 in the first and second PCR, respectively. Index (barcode) sequences were amplified using a Nextera XT Index Kit v2 setA (Illumina, San Diego, CA, USA) after Tag PCR amplification. The sequencing was performed using the paired–end Illumina MiSeq platform (2 × 300 base pairs [bp]). Subsequently, data processing, data assignment, and data aggregation were performed automatically using the originally created repertoire analysis software by Repertoire Genesis Inc. TCR (TRA and TRB) sequences were mapped to a reference sequence dataset from the international ImMunoGeneTics information system (IMGT) database (http://www.imgt.org, accessed on 1 November 2022) [30]. Nucleotide sequences of CDR3 regions ranged from a conserved cysteine at position 104 (Cys104) to a conserved phenylalanine at position 118 (Phe118), and the following glycine (Gly119) was translated into an amino acid sequence. A unique sequence read was defined as a sequence read with no identity in TRAV, TRAJ, and the deduced amino acid sequence of CDR3. The copy number of identical unique sequence reads in each sample was automatically counted and then ranked by copy number using software for repertoire analysis. The percentage occurrence frequencies of sequence reads containing TRAV and TRAJ, as well as genes, were calculated.

### 4.10. Statistical Analysis

Statistically significant differences between the mean values of each experimental group were analyzed using the Kruskal–Wallis test followed by Steel–Dwass’ multiple comparison tests and Mann–Whitney U test. All analyses were performed with EZR (Saitama Medical Center, Jichi Medical University, Saitama, Japan) [31], which is a graphical user interface for R (The R Foundation for Statistical Computing, Vienna, Austria). A *p*–value of <0.05 was considered significant, and a *p*–value of <0.01 was considered highly significant.

## Figures and Tables

**Figure 1 ijms-24-03137-f001:**
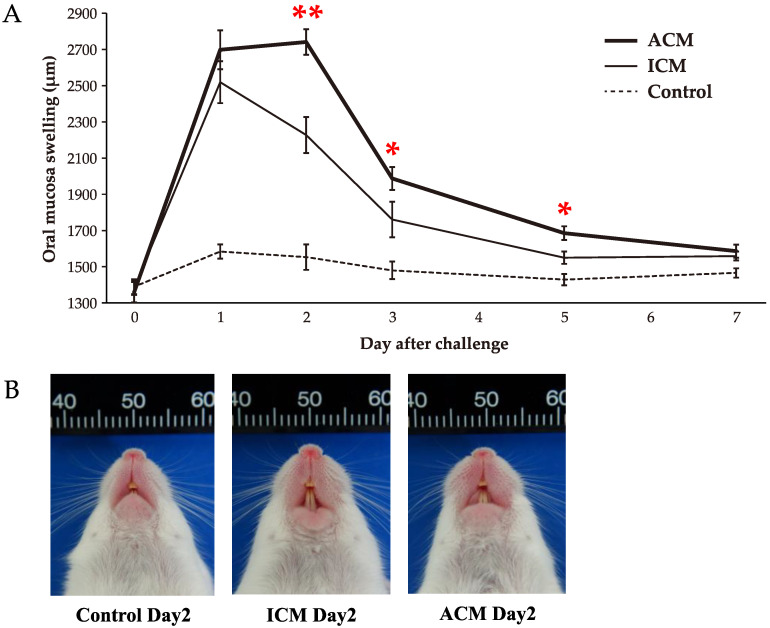
Swelling of the OM in Pd–induced allergic mice. (**A**) OM buccal area swelling from day 1 to day 7 after the challenge was measured in Pd–induced ACM and ICM and control mice (*n* = 10). Each point and error bars indicate the mean values and standard deviations. Statistical significance was tested through the Kruskal–Wallis test followed by Steel–Dwass’s multiple comparison tests (* *p* < 0.05, ** *p* < 0.01). (**B**) Macroscopic findings of the oral mucosa in Pd–induced allergic mice.

**Figure 2 ijms-24-03137-f002:**
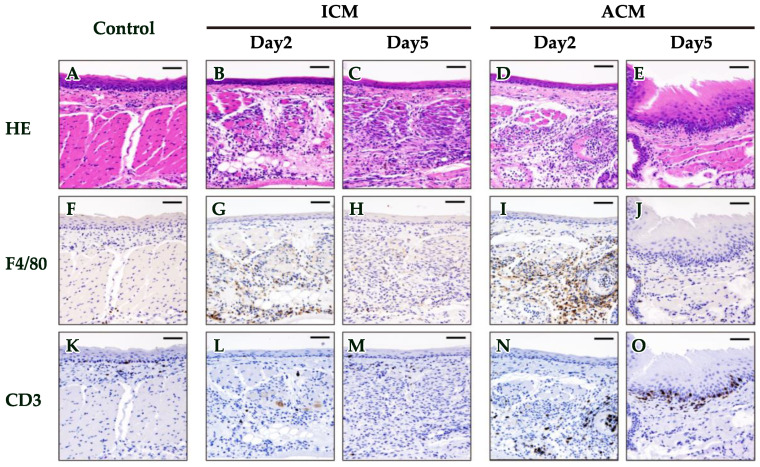
Histopathological and IHC analyses of F4/80 and CD3 in the OM of Pd–induced allergic mice. Representative photomicrographs of OM sections from control and Pd–induced ICM and ACM mice from days 2 and 5 after the challenge, stained with HE (**A**–**E**), F4/80 (**F**–**J**), and CD3 (**K**–**O**). Scale bar = 5 μm.

**Figure 3 ijms-24-03137-f003:**
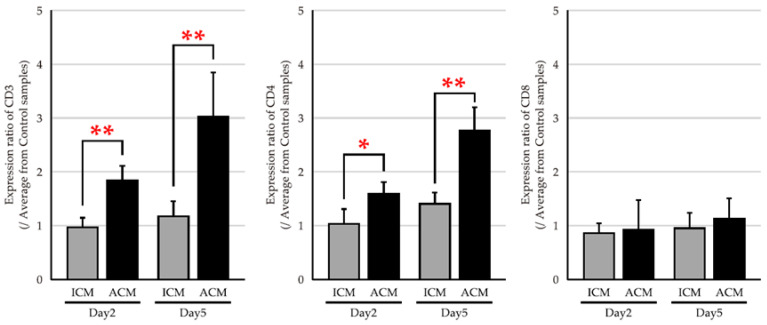
mRNA expression levels of T cell phenotypes in the OM of Pd–induced allergic mice. The mRNA expression levels of CD3, CD4, and CD8 were assessed in the OM of ICM (gray bars) and ACM (black bars) mice (*n* = 5) on days 2 and 5 after the challenge. GAPDH gene expression was used as an internal control. Each sample was divided by the average of the control sample to calculate the expression ratio *(n* = 5). Bars and error bars indicate the mean plus standard deviation. Statistical significance was tested by the Mann–Whitney U test (* *p* < 0.05, ** *p* < 0.01).

**Figure 4 ijms-24-03137-f004:**
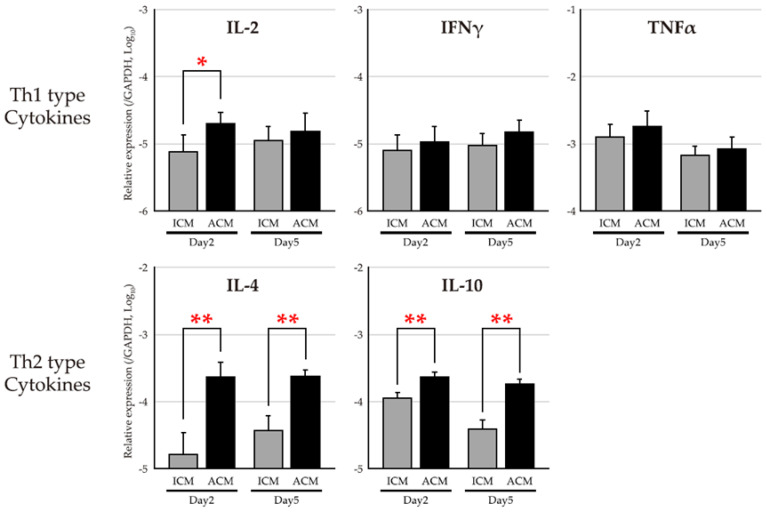
mRNA expression levels of Th1– and Th2–type cytokines in the mucosa of Pd–induced allergic mice. The mRNA expression levels of Th1–type cytokines (IL–2, IFNγ, and TNFα) and Th2 type cytokines (IL–4 and IL–10) were assessed in the OM of ICM (gray bars) and ACM (black bars) mice (*n* = 5) on days 2 and 5 after the challenge. Each sample was divided by GAPDH gene expression as an internal control. Bars and error bars indicate the mean plus standard deviation. Statistical significance was tested by the Mann–Whitney U test (* *p* < 0.05, ** *p* < 0.01).

**Figure 5 ijms-24-03137-f005:**
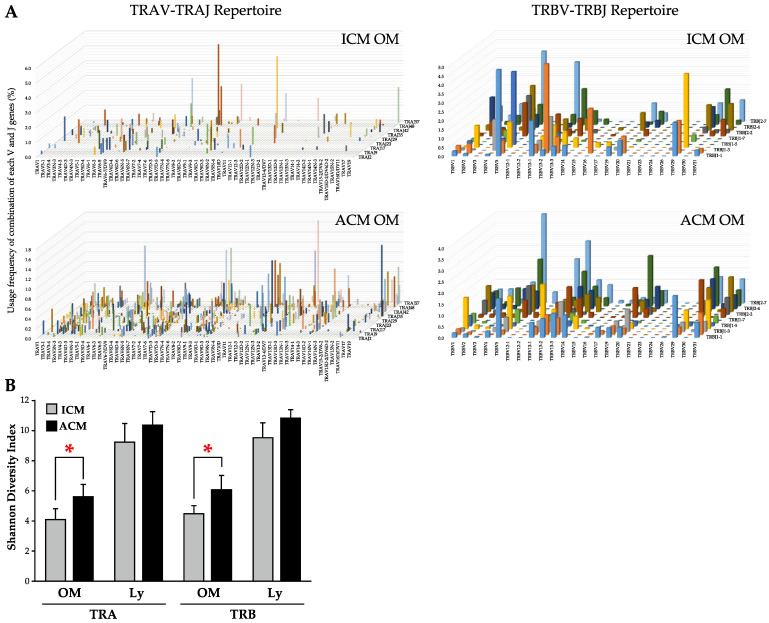
TCR repertoire analysis of the TRA and TRB in ICM and ACM mice. The NGS–based TCR repertoire analysis was performed on the OM and Ly of ICM and ACM mice on day 5 following the challenge. (**A**) Combining TRAV or TRBV on the *X*–axis and TRAJ or TRBJ on the *Z*–axis, with the frequency (percentage) of each clone on the *Y*–axis, 3D images depict the TCR repertoire (*n* = 3, average). (**B**) Shannon Diversity Index is shown in the ACM and ICM, OM and Ly, and TRA and TRB (*n* = 3). Statistical significance was tested by the Mann–Whitney U test (* *p* < 0.05).

**Figure 6 ijms-24-03137-f006:**
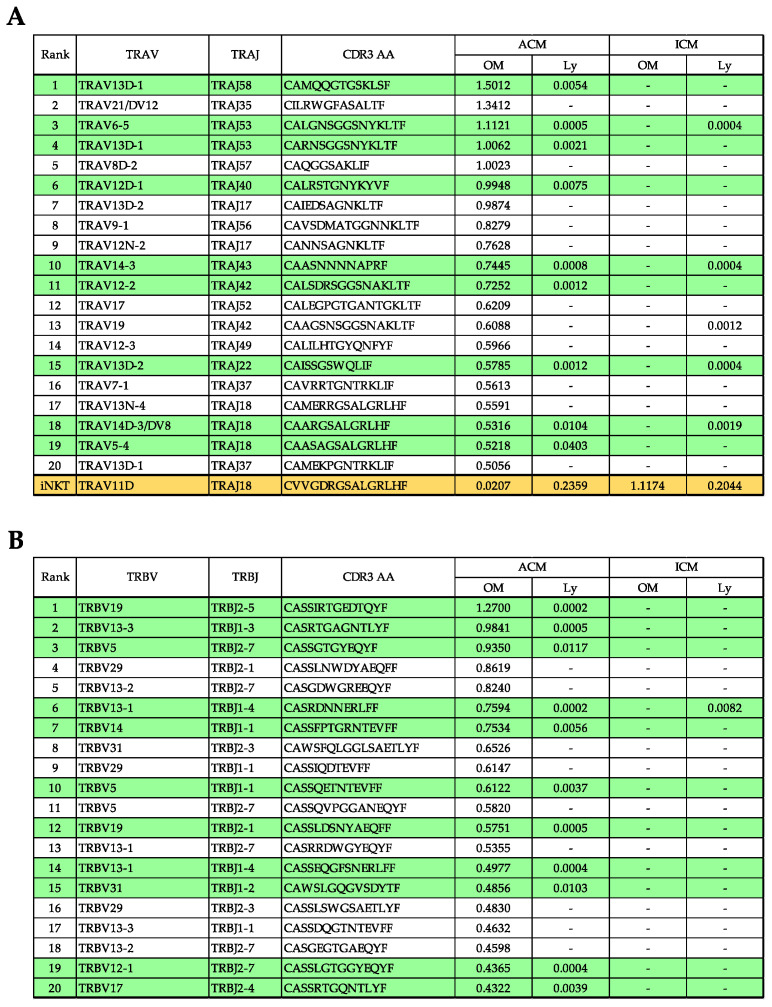
Commonality of unique reads obtained from the repertoire analysis in TRA and TRB. (**A**,**B**) The %frequency of unique reads in the top 20 in ACM OM mice and their commonality in other samples. The %frequency is the average of the frequency of presence for in–frame reads for each of the three independent samples. The green shading is unique reads shared by ACM OM and ACM Ly mice. (**A**,**B**) Orange shading included iNKT cell sequences as detected in all samples. (**A**) iNKT cell sequence and %frequency of presence in each sample. The %frequency is the average of the frequency of presence for in–frame reads for each of the three independent samples.

**Figure 7 ijms-24-03137-f007:**
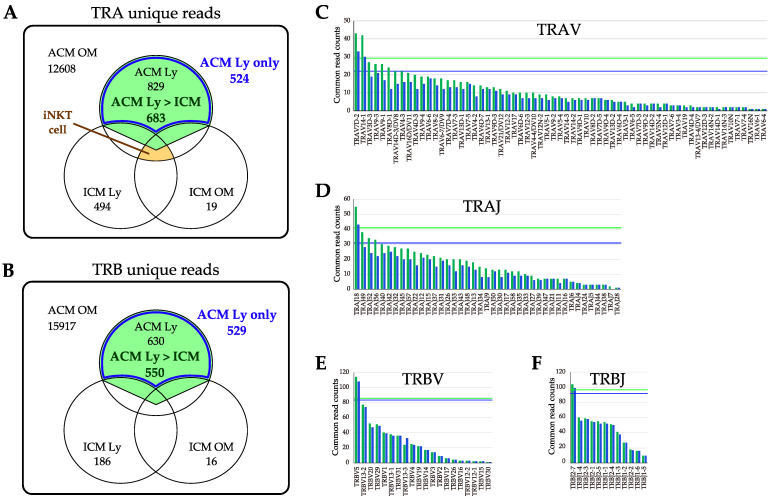
Common read counts of ACM OM and ACM Ly repertoire. (**A**,**B**) The number of unique reads common to the ACM OM mice and other samples is shown in the Venn diagram (*n* = 3, total). The green shading is unique reads present in ACM OM to those that were increased in ACM Ly mice over ICM. Blue box lines are unique reads present in ACM OM to those that were present only in ACM Ly mice. (**C**–**F**) The common unique reads between ACM OM and Ly mice were counted for TRAV, TRAJ, TRBV, and TRBJ genes, respectively. The green bars are unique reads present in ACM OM to those that were increased in ACM Ly mice over ICM. The blue bars are unique reads present in ACM OM to those that were present only in ACM Ly mice. The green line is the mean + 2SD of green bars. The blue line is the mean + 2SD of blue bars.

## Data Availability

This data presented in this study are available on request from the corresponding author.

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
