# Peer review of "Type IVb Hypersensitivity Reaction in the Novel Murine Model of Palladium–Induced Intraoral Allergic Contact Mucositis"

_ijms, 2023, doi:10.3390/ijms24043137_

Round 1

Reviewer 1 Report

In this study, the authors established novel Palladium (Pd) allergic mucositis model in mice. In this model, the authors demonstrated that CD4 T cells preferentially plays role in the pathogenesis. So far, it has been reported the murine model of Pd allergic contact dermatitis in foot or ear. To study Pd induced allergic contact dermatitis in dentistry, current study will be useful tool. However, some revisions of the manuscript are needed to improve this research as follow.

Major points

1) The authors should describe how to determine the concentration of PdCl2 to induce ACD. The authors denoted A strong relationship was found between exposure to dental alloy and ACD; thus, dental crowns seem to play a key role (page 2). What is the concentration of Pd used in dentistry reagents?

2) The peak of swelling in ACM was day 2 and swelling was gradually recovered (Figure 1). Contrary to this, T cell accumulation was increased up to day 5 (Figure 2, 3). These data suspected the significance of the relationship between T cell infiltration and OM swelling. I recommend that the authors should describe this discrepancy.

3) In Figure 1, to easily understanding for readers, I recommend the author to attach photograph of oral mucosa of ACM, ICM, and control mice beside graph. In this case, readers will be easy to decide which side is measure in buccal mucosa.

4) The authors demonstrated macrophage accumulation in ACM as compared with ICM (Figure 2). However, the authors did not describe the role of macrophage in the pathogenesis of oral mucositis in section of Introduction or Discussion. I think this point will make clearer to compare with the standard procedure of Pd induced allergic dermatitis in foot or ear.

5) Shannon diversity index of ACM was higher than that of ICM in oral mucosa (Figure 5B). I think the general inflammatory response will shrink TCR repertoire due to clonal expansion of ACM-responsive T cells. To clearer understand T cell response in ACM, the authors should explain more detail about this data.

6) It seems be difficult to understand the data in Figure 7. I recommend to describe the purpose of this analysis. In addition, I think that comparing the results between common reads of ACD in foot/ear and that of ACM will be helpful for demonstration of the novelty of this allergic model.

Minor point

In Materials and Methods section (4.8. qPCR), the authors described examination of transcription factors of regulatory T cells (p11). However, In Result section, the data was omitted. Thus, the authors should delete this sentence or add the result of expression of transcription factors

Reviewer 2 Report

The manuscript is interesting and well-written. The authors presented the establishment of a murine model of palladium-induced allergy including the assessment of cytokine profile and characterization of T cell receptor repertoire. The methods and results are described in an appropriate manner. Conclusions are based on the data obtained.

In my opinion, the Authors should:

1. explain in the Results section why they performed the analyses on day 2 and day 5.

2. change the labeling of statistical significance in Figure 5B to be the same as in the previous figures

Reviewer 3 Report

Nasu and Kumagai et al. developed a mouse model of delayed-type hypersensiticity (DTR) induced by palladium (Pd), which is a common component of dental alloy.

This study follow the group’s previous work (Ref. 9).

  The outcome of allergic responses against haptens depends on their chemical nature, which presumably male DTR of different profile compared with the nickel (Ni)-DTR.

Here are the comments.

Major

1. The reviewer agrees to the authors’s claim that they established a Pd-DTR model; however, the reviewer still feels the rationale of publishing data using very similar experimental setting as the ref. 9. The authors should describe Pd’s features distinctive from Ni, from clinical or biochemical perspectives.

2. It should be theoretically possible that elicitation can be achieved without adding lipopolysaccharide (LPS), which should circumvent the early, non-specific phase. The authors could test this setting, if they have not. 

3. Cd1d–/– mice could have been used to reverse the phenotype, like ref. 9.

Round 2

Reviewer 1 Report

The authors improved manuscript and respond my comments sincerely. These comments are almost appropriate and I recommend that this manuscript will be accepted.

Reviewer 3 Report

No comments.